

# YOLOv8s-CGF: a lightweight model for wheat ear Fusarium head blight detection

Chengkai Yang[1], Xiaoyun Sun[1], Jian Wang[1], Haiyan Lv[1], Ping Dong[1], Lei Xi[1] and Lei Shi[1,2]

[1] College of Information and Management Science, Henan Agricultural University, Zhengzhou, Henan, China
[2] Henan Grain Crop Collaborative Innovation Center, Henan Agricultural University, Zhengzhou, Henan, China

## ABSTRACT

Fusarium head blight (FHB) is a destructive disease that affects wheat production. Detecting FHB accurately and rapidly is crucial for improving wheat yield. Traditional models are difficult to apply to mobile devices due to large parameters, high computation, and resource requirements. Therefore, this article proposes a lightweight detection method based on an improved YOLOv8s to facilitate the rapid deployment of the model on mobile terminals and improve the detection efficiency of wheat FHB. The proposed method introduced a C-FasterNet module, which replaced the C2f module in the backbone network. It helps reduce the number of parameters and the computational volume of the model. Additionally, the Conv in the backbone network is replaced with GhostConv, further reducing parameters and computation without significantly affecting detection accuracy. Thirdly, the introduction of the Focal CIoU loss function reduces the impact of sample imbalance on the detection results and accelerates the model convergence. Lastly, the large target detection head was removed from the model for lightweight. The experimental results show that the size of the improved model (YOLOv8s-CGF) is only 11.7 M, which accounts for 52.0% of the original model (YOLOv8s). The number of parameters is only $5.7 \times 10^6$ M, equivalent to 51.4% of the original model. The computational volume is only 21.1 GFLOPs, representing 74.3% of the original model. Moreover, the mean average precision (mAP@0.5) of the model is 99.492%, which is 0.003% higher than the original model, and the mAP@0.5:0.95 is 0.269% higher than the original model. Compared to other YOLO models, the improved lightweight model not only achieved the highest detection precision but also significantly reduced the number of parameters and model size. This provides a valuable reference for FHB detection in wheat ears and deployment on mobile terminals in field environments.

## INTRODUCTION

Wheat, as one of the three major grain crops in China, has always been a focus of attention for those concerned with its yield and quality (*Wang, Li & Su, 2023*). Wheat Fusarium head blight (FHB) is a type of wheat disease that affects a vast global area. It not only causes a reduction in grain production but also poses a threat to the health of humans and animals (*Ochodzki et al., 2021*; *Femenias et al., 2020*). Therefore, timely and accurate

Corresponding author
Lei Shi, shilei@henau.edu.cn

identification of wheat FHB can provide an essential guarantee for improving wheat yield and quality.

With the development of artificial intelligence, machine learning-based target detection methods are beginning to find applications in agriculture (*Xu et al., 2021*; *Zhang, Zou & Pan, 2020*). *Khan et al. (2021)* proposed a machine learning-based early detection model for wheat powdery mildew hyperspectral images, achieving an overall accuracy of over 82.35%. *Zhang, Zou & Pan (2020)* utilized an apple fruit segmentation algorithm and developed nine different machine learning algorithm-based classifiers, effectively segmenting apple fruits in orchard images. *Huang et al. (2021)* established a red mold detection method based on continuous wavelet analysis and particle swarm optimization support vector machine (PSO-SVM), achieving a detection accuracy of 93.5%.

Although crop disease detection based on traditional machine learning algorithms can achieve an ideal recognition effect, these methods for the target detection of crops are easily influenced by factors such as terrain, weather, and the distance between the camera and crops. This makes it challenging to obtain accurate recognition in complex situations. In recent years, deep learning methods have gradually been applied to the field of agricultural production (*Díaz-Martínez et al., 2023*). *Mi et al. (2020)* constructed a deep learning network, C-DenseNet, incorporating the convolutional block attention module (CBAM) attention mechanism to classify wheat stripe rust. The results showed that the test accuracy of C-DenseNet reached 97.99%. *Wang et al. (2023)* used an algorithm based on improved YOLOv5s for recognizing corn and weeds in the field. The results indicated that the AP value of corn reached 96.3%, and the AP value of weeds reached 88.9%. *Yang et al. (2023a)* improved the YOLOv7 algorithm to identify fruits. The improved YOLOv7 algorithm achieved an accuracy rate of 96.7% on the test set. *Ma et al. (2023)* proposed an improved YOLOv8 algorithm for the detection of wheat stripe rust, leaf rust, and powdery mildew. The experimental results indicate that the mAP of the improved YOLOv8 model for detecting the three wheat leaf diseases is 98.8%.

Traditional identification of wheat FHB primarily relies on manual detection. With the development of computer vision, deep learning algorithms are now being applied to wheat FHB detection. *Su et al. (2021)* proposed an algorithm based on the improved Mask-RCNN network to assess wheat FHB severity, achieving a prediction accuracy of 77.19%. *Hong et al. (2022)* utilized a lightweight improved YOLOv4-based FHB detection model for wheat ears, attaining an accuracy of 93.69% in wheat FHB detection. *Zhang et al. (2022a)* improved the YOLOv5 target detection network for wheat FHB detection, and the results demonstrated an average detection accuracy of 90.67%, which could satisfy the identification of wheat FHB under field conditions.

Although the above recognition of wheat FHB has achieved good results, either the recognition accuracy is not high, or is not conducive to the deployment of mobile devices. This article addresses the shortcomings of current target detection models in identifying wheat FHB, such as low accuracy, large parameters and high computation, increased model size leading to higher memory usage, and unsuitability for mobile terminals. The focus of this study is to create a lightweight model for identifying wheat FHB, providing

valuable references for identifying FHB in field environments and deploying it on mobile terminals.

# MATERIALS AND METHODS

## Winter wheat FHB image dataset construction

The wheat FHB images were collected in June 2022 at the Rocky Ford FHB nursery, Kansas State University. Four winter wheat varieties "Clark", "Jagger", "Overley" and "Everest" were used as the plant materials. The image acquisition device was a high-pixel smartphone, and the shooting process involved top-down photography. The resolution of the images was $3,024 \times 4,032$, with a total of 231 images. Crop images with a resolution of $832 \times 832$ from each original image. In this study, data augmentation on the self-built dataset was expanded to 1,386 samples, randomly divided into a training set and a validation set at the ratio of 9:1. An example of wheat FHB images is shown in Fig. 1.

The images were labeled using the LabelImg tool to generate the corresponding XML files. These files contained location information for both healthy and diseased wheat ears, image size, and their respective category information. Mosaic data augmentation was used to handle data samples during training process. This involved randomly cropping and splicing four images into one image as training data to enrich the image background and enhance the model training efficiency.

## YOLOv8 model

YOLOv8 is a target detection model and the latest version in the YOLO series of models (*Redmon & Farhadi, 2018*; *Bochkovskiy, Wang & Liao, 2020*; *Zou et al., 2022*). It offers improved speed and accuracy compared to previous YOLO models. YOLOv8 consists of five versions: YOLOv8n, YOLOv8s, YOLOv8m, YOLOv8l, and YOLOv8x. These versions are arranged based on model size, with YOLOv8n being the smallest and YOLOv8x being the largest. Since this study focuses on recognizing two categories, namely healthy wheat ears and diseased ears, and considers the need for real-time detection and easy deployment, the YOLOv8s model is selected as the base model. This model has fewer parameters and requires less computational effort.

The YOLOv8s model is made up of four components: Input, Backbone, Neck, and Detection. The Input layer is used to pass the image into the model and perform preprocessing operations on it, while the Backbone network for extracting image features. The Neck layer performs feature fusion on the extracted features, and the Detection layer predicts features of three different dimensions, obtaining category and location information predicted by the network. In this study, the YOLOv8s model served as the base for identifying healthy wheat ears and FHB ears. Additionally, lightweight improvements were made to make it more suitable for deployment on mobile devices.

## YOLOv8s-CGF model

Since the FasterNet (*Chen et al., 2023*) network can effectively reduce model computational redundancy and memory access, improving model speed without affecting accuracy, this article proposes the C-FasterNet module based on the FasterNet module. It

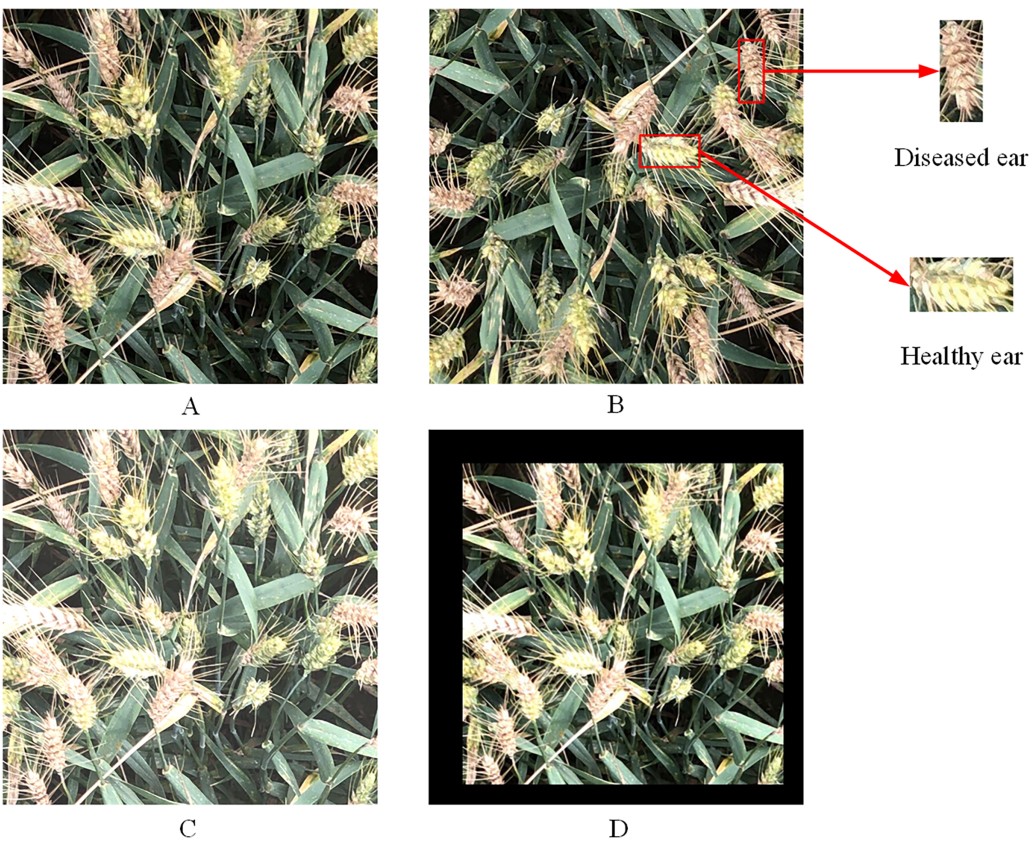

**Figure 1 Images of wheat FHB.** (A) Original image; (B) rotated image; (C) image after adjusting brightness; (D) scaled image.

serves as the module for the main learning of features in the backbone network of YOLOv8s, which can be used to improve the model speed without affecting accuracy, while reducing the number of parameters and computation. GhostConv (*Han et al., 2020*) can generate more feature maps from inexpensive operations, fully revealing feature information. Therefore, replacing Conv in the backbone network with GhostConv reduces the model size while improving a certain degree of accuracy. Subsequently, the loss function is replaced with Focal CIoU to speed up model convergence. Finally, the large target detection head in the network is removed for further lightweight. The structure of the YOLOv8s-CGF model is shown in Fig. 2.

### C-FasterNet network

C-FasterNet is a proposed module based on FasterNet. The FasterNet network is a new fast neural network proposed in 2023. The design of PConv (partial convolution) in the network exploits the redundancy in the feature maps by applying regular convolution only to some of the input channels, leaving the others unchanged. For consecutive or regular memory accesses, the first or last consecutive channel is considered a representative of the whole feature map for computation. Figure 3 shows how PConv works.

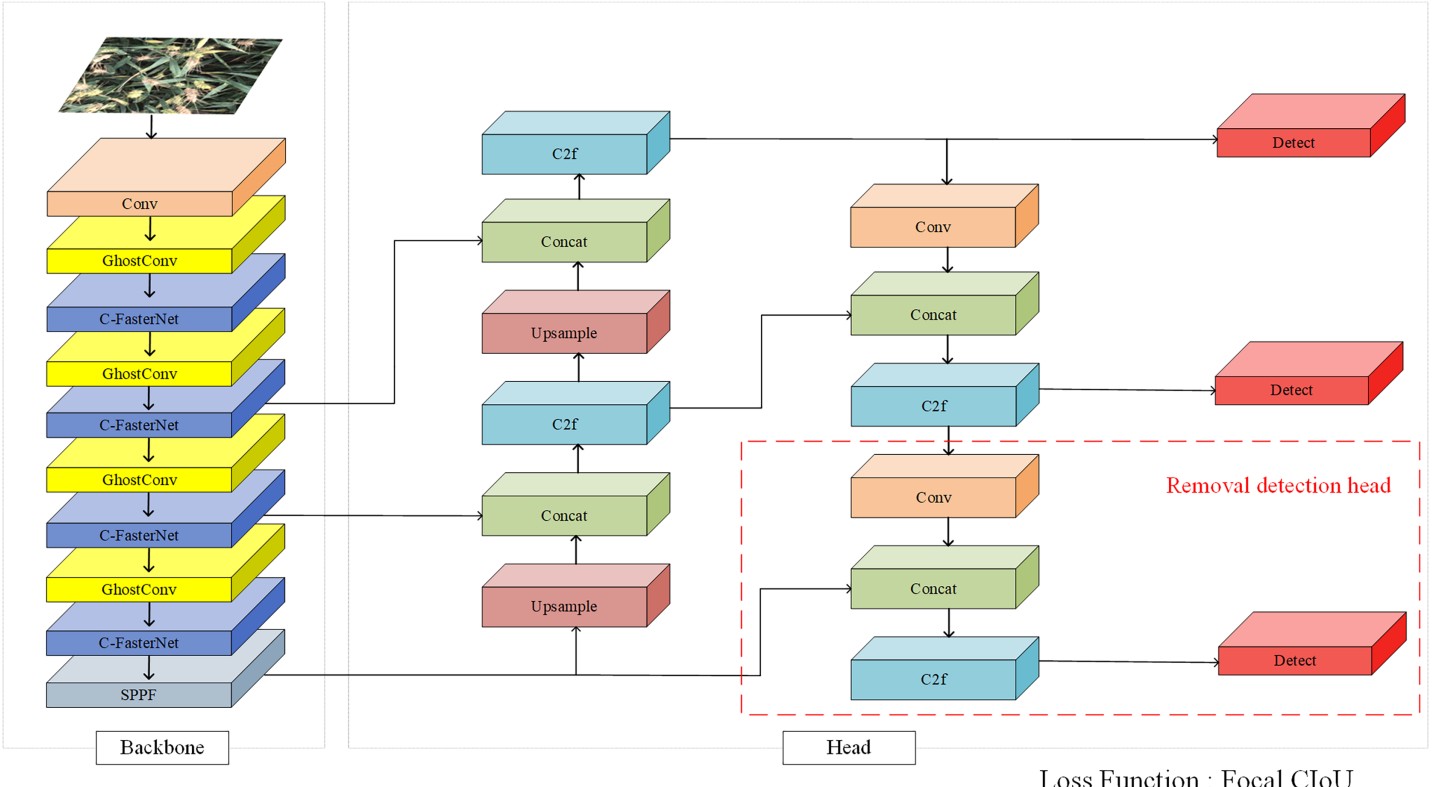

**Figure 2 YOLOv8s-CGF model.**

**Figure 3 PConv working principle diagram.**

Figure 4 shows the whole structure of FasterNet, which consists of four stages. Each stage is preceded by an embedding stage (a regular Conv 4 × 4 with a step size of 4) or a merging stage (a regular Conv 2 × 2 with a step size of 2) for spatial downsampling and channel count expansion. Each stage has a bunch of FasterNet blocks, and the blocks in the last two stages consume fewer memory accesses while having higher floating point operations per second (FLOPS). Therefore, more FasterNet blocks are placed in the last two stages, and accordingly, more computation is allocated to these stages. The last three
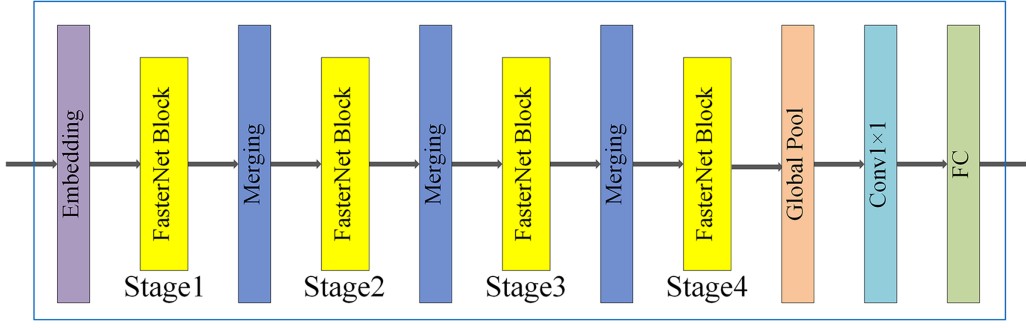

**Figure 4 FasterNet network structure.**

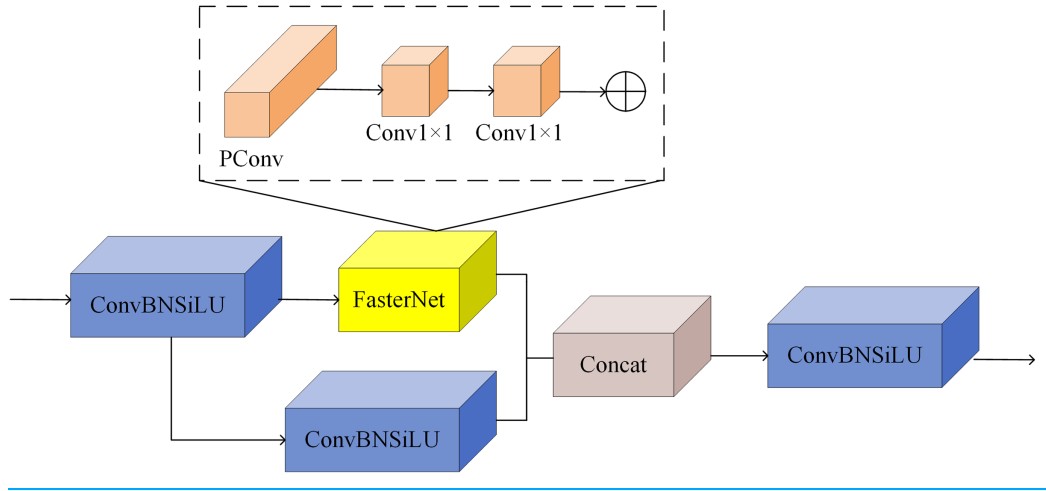

**Figure 5 C-FasterNet model.**

layers include global average pooling, Conv 1 × 1, and a fully connected layer for feature transformation and classification.

In this study, based on the above FasterNet module, we propose a C-FasterNet module, which consists of three standard convolutional layers and several FasterNet modules. C-FasterNet is the module that performs the main learning of the residual features. Its structure is divided into two branches: one that uses a stack of several FasterNet modules and a standard convolutional layer, and the other that only uses one standard convolution. Finally, the two branches are subjected to a Concat operation. The C-FasterNet module is used for YOLOv8s network feature extraction, reducing redundant computation and memory access while efficiently extracting spatial features. Its structure is shown in Fig. 5.

### GhostConv structure

GhostNet is a lightweight CNN network proposed by the Huawei team. Its core component, the Ghost module (GhostConv), generates a portion of feature maps through original convolution, while the rest is generated using a Cheap operation. This Cheap operation can be a linear transformation of the remaining feature map or a similar feature map generated by Depthwise convolution on the output of the original convolution. The structure of GhostConv is shown in Fig. 6, where C1 and C2 represent the input and

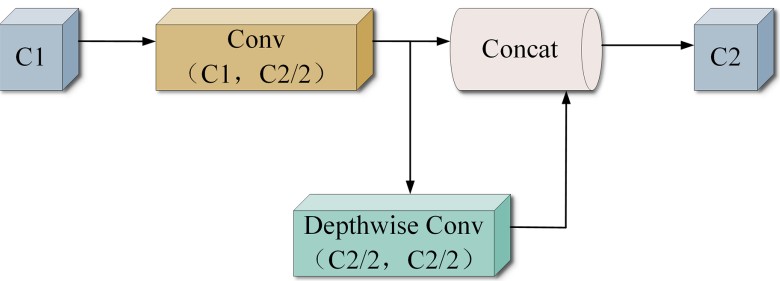

**Figure 6 GhostConv structure.**

output channels, respectively. Half of the output feature maps come from one regular convolution, while the other half is generated by a 5 × 5 Depthwise convolution on the result of the first one. GhostConv achieves the same or even more efficient feature extraction with less complexity than the original convolutional layer. Whereas in CNN networks there is generally a lot of redundant convolutional computation with intermediate feature maps, Ghost convolution forces the network to learn useful features from half of the convolutional kernels. Simultaneously, Depthwise convolution is utilized to generate a feature map for this convolutional portion. The use of a 5 × 5 Depthwise convolution helps widen the receptive field of the generated feature map, enhancing the overall information contained within it.

In this study, GhostConv is employed to replace the convolution module in YOLOv8s, aiming to compress the model rapidly and efficiently and extract target features.

### Focal CIoU loss function

There is often a serious imbalance between positive and negative samples in target detection. However, the default CIoU loss (*Zhang et al., 2022b*) of the YOLOv8s model, which is equal for all samples, does not effectively address this problem. As a result, the model will excessively focus on prediction boxes that have less overlap with the true value, leading to a degradation of model performance. To solve this problem, a Focal CIoU loss function is introduced. This function increases the contribution of positive samples in $L_{CIoU}$ by resetting the weights in $L_{CIoU}$ according to the IoU values. IoU measures the overlap between the predicted box (A) and the true box (B). The formula for Focal CIoU is as follows:

$$L_{\text{Focal\_CIOU}} = IoU^{\gamma} L_{\text{CIOU}}, \tag{1}$$

where IoU denotes intersection over union, the parameter $\gamma$ determines the degree of outlier suppression, and $L_{CIoU}$ denotes CIoU loss. $\gamma$ has a default value of 0.5.

$$IoU = \frac{|A \cap B|}{|A \cup B|}, \tag{2}$$

$$CIoU = 1 - IoU + \frac{\rho^2(b, b^{gt})}{c^2} + \alpha v, \tag{3}$$

where $b$ and $b^{gt}$ denote the centroids of the predicted bounding box and the true bounding box, respectively; $\rho$ denotes the Euclidean distance between the two centroids; $c$ denotes

the diagonal length of the smallest closed rectangle containing the predicted bounding box and the true bounding box; $v$ is used to quantify the consistency of the aspect ratio; and $\alpha$ is a weight function. The equations for $v$ and $\alpha$ are as follows:

$$v = \frac{4}{\pi^2}\left(\arctan\frac{w^{gt}}{h^{gt}} - \arctan\frac{w}{h}\right), \tag{4}$$

$$\alpha = \frac{v}{1 - IoU} + v, \tag{5}$$

where $w^{gt}$ and $h^{gt}$ denote the width and height of the ground-truth bounding box, and $w$ and $h$ denote the width and height of the prediction bounding box, respectively.

## Experimental environment and parameters

All experiments in this study use PyTorch as the deep learning model framework, and the GPU is NVIDIA GeForce RTX3090. The optimizer chosen is the AdamW optimizer, with an initial learning rate of 0.01, momentum set to 0.937, weight decay at 0.0005. The input image size is $640 \times 640$, the batch size is set to 32, and the model is trained for 200 epochs.

## Model evaluation metrics

To evaluate the detection effect of the algorithm on wheat ears, the model performance is examined using recall, precision, average precision and mean average precision as evaluation metrics. The number of parameters, computation, and model size are used to reflect the complexity of the model. The recall reflects the model's ability to find positive sample targets, the precision reflects the model's ability to classify samples, and the average precision reflects the model's overall performance in target detection and classification. The calculation formula is as follows:

$$Precision = \frac{TP}{TP + FP}, \tag{6}$$

$$Recall = \frac{TP}{TP + FN}, \tag{7}$$

$$AP = \int_0^1 P \cdot RdR, \tag{8}$$

$$mAP = \frac{1}{N}\sum_{i=1}^{N} AP_i, \tag{9}$$

where $TP$ denotes the number of correctly identified positive samples, $FP$ denotes the number of incorrectly identified positive samples, $FN$ denotes the number of incorrectly identified negative samples and $N$ represents the number of categories of data. Positive and negative samples are judged by setting the average IoU threshold between the predicted target region and the actual target region, and if the IoU of both exceeds this threshold, it is a positive sample, and vice versa, it is a negative sample. mAP@0.5 is the AP evaluated by the object detection model with an IoU value of 0.5, and mAP@0.5 is the mean of all its classes. mAP@0.5:0.95 represents the average mAP over different IoU thresholds (from 0.5 to 0.95, step size 0.05) (0.5, 0.55, 0.6, 0.65, 0.7, 0.75, 0.8, 0.85, 0.9, 0.95).

**Table 1 Ablation experiment results.**

| Test number | C-FasterNet | Ghost Conv | Focal CIoU | Detection head | Parameters/ ×10⁶M | Computation/ GFLOPs | Model size/ MB | mAP@0.5/ % | mAP@0.5:0.95/ % |
|---|---|---|---|---|---|---|---|---|---|
| 1 | × | × | × | × | 11.1 | 28.4 | 22.5 | 99.489 | 92.515 |
| 2 | ✓ | × | × | × | 10.1 | 25.8 | 20.5 | 99.486 | 92.115 |
| 3 | × | ✓ | × | × | 10.4 | 26.6 | 21.0 | 99.500 | 92.722 |
| 4 | × | × | ✓ | × | 11.1 | 28.4 | 22.5 | 99.491 | 92.526 |
| 5 | × | × | × | ✓ | 7.5 | 25.5 | 15.2 | 99.492 | 92.079 |
| 6 | ✓ | ✓ | ✓ | × | 9.4 | 24.1 | 18.1 | 99.495 | 92.931 |
| 7 | ✓ | × | ✓ | ✓ | 6.5 | 22.9 | 13.2 | 99.488 | 92.210 |
| 8 | ✓ | ✓ | ✓ | ✓ | 5.7 | 21.1 | 11.7 | 99.492 | 92.784 |

# RESULTS AND ANALYSIS

## Results of ablation experiment

Aiming at the problems that the original YOLOv8s model has a large number of parameters and computations, and the model size takes up a large amount of memory, this study improves the structure of the YOLOv8s network by proposing the C-FasterNet module based on the FasterNet module as the main learning module of the YOLOv8s backbone network. Then, it replaces the Conv in the YOLOv8s backbone network with GhostConv to reduce model size and improve accuracy. GhostConv in the YOLOv8s backbone network to reduce the model size while improving accuracy. Introducing Focal CIoU reduces the loss value and accelerates the model convergence. Finally, removing the large target detection head in the network further reduces the size of the model without affecting the accuracy of the model, making it more suitable for the recognition of wheat FHB and later deployment. To analyze and validate the improved lightweight network model, eight groups of ablation experiment are designed, and the specific experiment results are shown in Table 1.

According to the ablation experiment in Table 1, it can be observed that using the C-FasterNet module to replace the C2f module in the model backbone network reduces the number of parameters and computation amount of the model, and also diminishes the size of the weights generated by the model, while basically keeping the mAP unchanged and slightly lowering the recall rate. The reason for this analysis is that the C-FasterNet module utilizes the FasterNet module, which reduces the number of memory accesses and the model's computational redundancy, and has higher FLOPS. This results in a reduction of the number of parameters in the model, improvement in model computation speed, and ensures the network is lightweight while maintaining high recognition accuracy. Additionally, from the ablation experiment, it can be observed that after replacing GhostConv, the number of parameters in the model and the computation amount are further reduced, and the model precision is slightly improved. The analyzed reason is that GhostConv can generate more feature maps from cheap operations. Based on a set of intrinsic feature maps, a series of linear transformations are applied at low cost to generate many reshaped feature maps, fully revealing the information of intrinsic features. With the

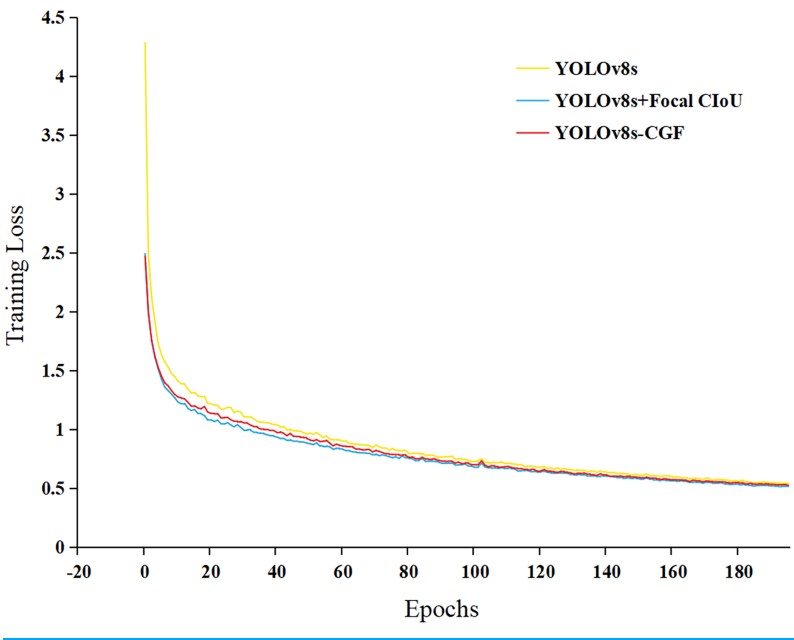

**Figure 7 Loss in training.**

introduction of Focal CIoU, the number of parameters and the amount of computation in the model did not change, and the mAP values remained essentially unchanged. Finally, the large target detection head in the network is removed to further reduce the number of model parameters and the weight size. Figure 7 shows the graph of the loss value changes of the model during the training process, indicating that the Focal CIoU loss function can achieve smaller loss values than CIoU, and the model convergence is faster. Combining the ablation experiment, the YOLOv8s-CGF model proposed in this study ensures high precision in wheat FHB identification and achieves the goal of lightweight.

## Comparative analysis of different models

To assess the performance of the improved lightweight network model with other algorithmic models and explore the superiority of the improved algorithms in this study, other target detection algorithms from the YOLO series, such as YOLOv5s, YOLOv6s, and YOLOv7-tiny, are selected for the comparative experiment. The variation of the mAP@0.5 curve for each model on the training dataset is depicted in Fig. 8.

As shown in the figure, mAP improves rapidly in the early stages of training. The model exhibits fluctuations between 10 and 60 epochs, but the overall trend is upward. After 80 epochs, mAP converges and stabilizes, with YOLOv8s-CGF achieving the highest mAP. The performance comparisons for each model are presented in Table 2.

According to the experimental results in Table 2, it can be seen that the YOLOv7-tiny model has the lowest mAP. This is attributed to the YOLOv7-tiny model having fewer parameters and less computation, which hinders achieve higher detection precision. The YOLOv6s model has the highest number of parameters, largest computation and model weight, which does not meet the requirement for lightweight. While the YOLOv5s model achieves good precision with fewer parameters and computation, its precision, recall and

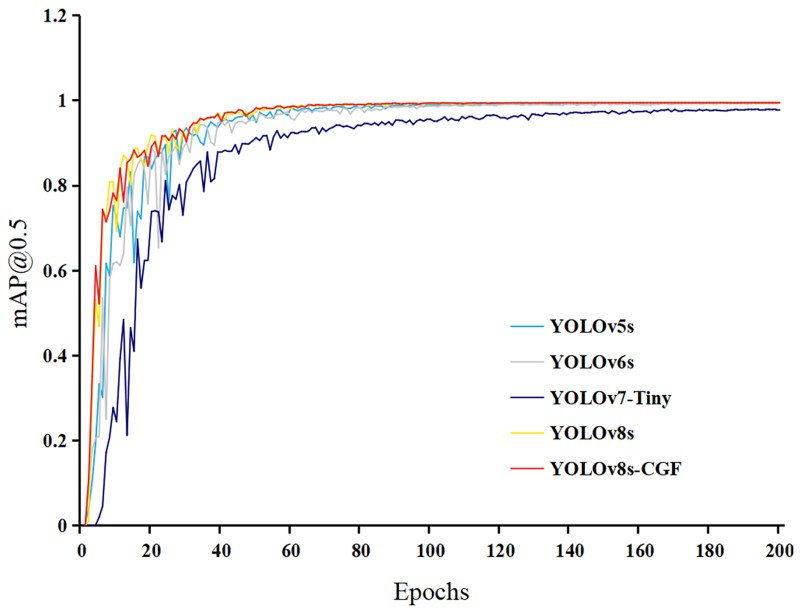

**Figure 8 Each model mAP@0.5 curve.**

**Table 2 Compare experiment results with different models.**

| Model name | Parameters/×10⁶M | Computation/GFLOPs | Model size/MB | Precision | Recall | mAP@0.5 | mAP@0.5:0.95 |
|---|---|---|---|---|---|---|---|
| YOLOv5s | 7.0 | 15.8 | 14.4 | 0.992 | 0.985 | 0.994 | 0.841 |
| YOLOv6s | 16.3 | 44.0 | 32.8 | 0.978 | 0.98 | 0.993 | 0.875 |
| YOLOv7-Tiny | 6.0 | 13.2 | 12.3 | 0.995 | 0.927 | 0.977 | 0.708 |
| YOLOv8s | 11.1 | 28.4 | 22.5 | 0.994 | 0.997 | 0.995 | 0.925 |
| YOLOv8s-CGF | 5.7 | 21.1 | 11.7 | 0.996 | 0.996 | 0.995 | 0.928 |

mAP are slightly lower compared to the YOLOv8s model. The YOLOv8s-CGF model proposed in this study has the fewest parameters, generates the smallest weight file, and attains the highest precision, recall, and mAP. It achieves superior recognition precision while remaining lightweight, and its detection speed is higher, meeting real-time detection requirements. The recognition results of different models on wheat FHB are shown in Fig. 9.

## Statistical analysis

To analyze whether the performance improvement of the modified model is significantly enhanced from a statistical perspective, 10 experiments were conducted for both the pre-improvement and post-improvement models using different random seeds. Paired samples t-test were employed to analyze six performance metrics, and the results are presented in Table 3.

From the table, it is evident that there is a significant difference among groups in terms of the precision metric ($P < 0.05$). Therefore, it is considered that the improvement method

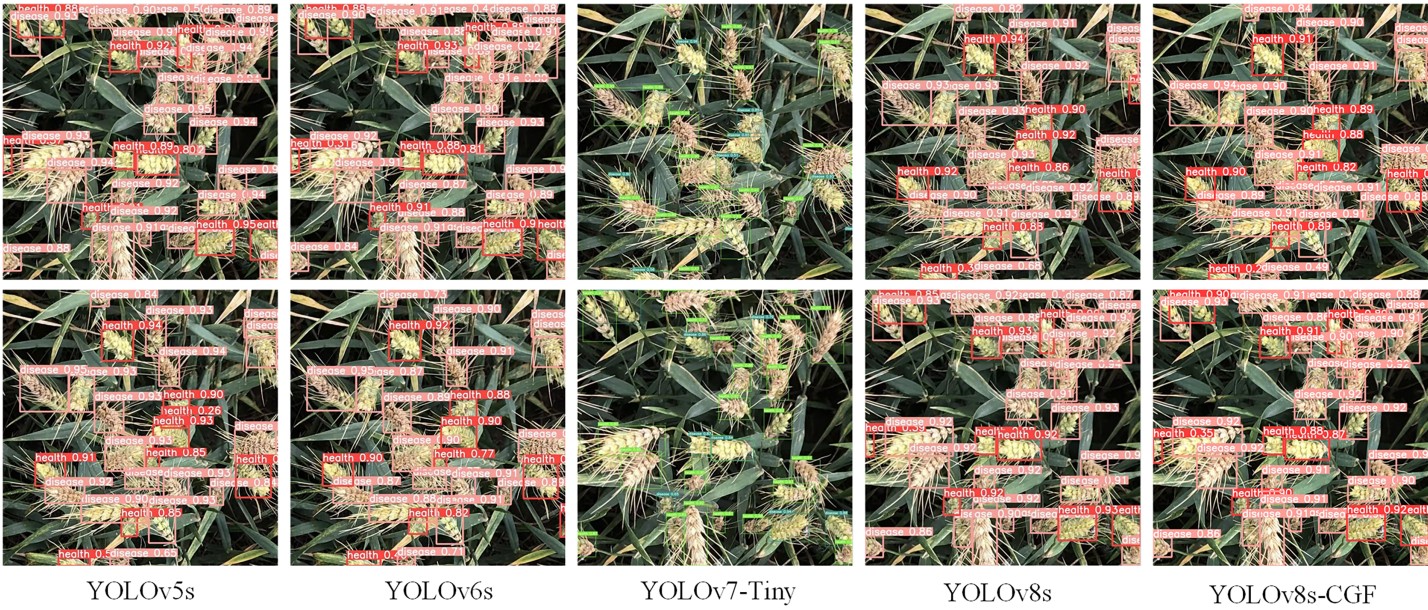

| YOLOv5s | YOLOv6s | YOLOv7-Tiny | YOLOv8s | YOLOv8s-CGF |

**Figure 9** Prediction results for different models.

**Table 3 Statistical test.**

| Model name | Parameters | Computation | Model size | Precision | Recall | mAP@0.5:0.95 |
|---|---|---|---|---|---|---|
| T | – | – | – | −3.122 | 1.309 | −3.444 |
| P | 0.000 | 0.000 | 0.000 | 0.012 | 0.222 | 0.007 |

has a significant impact on precision (*i.e.*, significantly improving the model's precision). On the recall metric, there is no significant difference among groups ($P > 0.05$), indicating that the improvement method has no significant impact on recall (*i.e.*, cannot significantly improve the model's recall). Regarding the mAP@0.5:0.95 metric, there is a significant difference among groups ($P < 0.05$), suggesting that the improvement method has a significant impact on mAP@0.5:0.95 (*i.e.*, significantly improving the model's mAP@0.5:0.95). Due to the fixed parameters, computation, and model size before and after the model improvement, and the significant reduction in parameters, computation, and model size after the improvement, it is considered that the improved model exhibits significance in these three metrics compared to the original model. In summary, the improved model shows significant differences in all five metrics, indicating that the proposed improvement method in this study significantly enhances model performance.

## DISCUSSION

At present, machine learning-based methods have been applied to the field of agricultural disease recognition (*Basavaiah & Arlene Anthony, 2020*; *Chemchem, Alin & Krajecki, 2019*; *Kayad et al., 2019*). Although relatively good results have been achieved, the recognition

processes are cumbersome. With the development of artificial intelligence, researchers have gradually turned to deep convolutional neural networks for the classification and recognition of crop diseases, and have achieved better results than traditional machine learning algorithms in crop identification (*Liu et al., 2020*; *Zhang, Kang & Wang, 2022*), weed identification (*Gallo et al., 2023*; *Tripathi, Yadav & Rai, 2022*), and pest identification (*Yang et al., 2023b*; *Talukder et al., 2023*). In recent years, an increasing number of scholars have focused on developing lightweight models by reducing parameters, computation, and model size, aiming for convenient deployment on mobile terminals. *Cong et al. (2023)* introduced a lightweight mushroom detection model, MYOLO, exhibiting a 2.04% increase in mAP and a 2.08-fold reduction in the number of parameters. This model lays a crucial theoretical foundation for the automated harvesting of fresh shiitake mushrooms. *Fang, Zhen & Li (2023)* devised a lightweight multi-scale convolutional neural networks (CNN) model, integrating the residual module and the inception module to recognize six wheat diseases, achieving an impressive 98.7% accuracy on the test dataset. Furthermore, *Jia et al. (2023)* enhanced the YOLOv7 algorithm for identifying rice pests and diseases. Utilizing the lightweight network MobileNetV3 as the backbone, this approach reduced the number of model parameters and combined attention mechanisms (CA) with the Scylla Intersection over Union (SIoU) loss function to enhance accuracy, resulting in a remarkable mAP@0.5 of 93.7% for the lightweight model.

In recent years, techniques utilizing hyperspectral and multispectral imaging have shown promise in wheat FHB detection in the field (*Mustafa et al., 2022*). *Gao et al. (2023)* proposed a UAV multispectral method that combines Vegetation Indices (VIs), Texture Indices (TIs), and an XGBoost model for FHB monitoring. The results demonstrated an accuracy of 93.63% on the test set. *Ma et al. (2021)* employed a wheat FHB detection model based on spectral feature combination, revealing that the combination of spectral bands, vegetation indices, and wavelet features performed exceptionally well in wheat FHB detection. However, these techniques involve complex steps and expensive instruments (*Bernardes et al., 2022*). In contrast, deep learning models offer promising applications in crop disease detection, allowing high-precision detection with low complexity by training on ordinary RGB images. Hence, this study proposes a target detection model for wheat FHB based on an improved YOLOv8s. This model not only features the fewest parameters and computations, resulting in the smallest model size, but also excels in recognizing wheat FHB. It achieves the highest mAP without issues such as detection gaps or leaks, ensuring complete coverage of wheat FHB within the detection frame. The model is designed for future deployment on mobile devices or embedding in UAVs for wheat FHB disease monitoring, enhancing the model's applicability in real-world scenarios.

Although the model performs well in recognizing wheat FHB, it currently can not classify disease severity and be applied to all wheat field. In future work, diseased wheat ear images of more varieties and growth stages will be added to enrich the dataset and enhance the generalization ability of the model. In addition, counting the number of diseased wheat ears and calculating the disease ear rate will be used to grade the disease severity, it will be beneficial for farmers to carry out scientific prevention and appropriate treatment

strategies. Furthermore, improving the different models to make them more lightweight and practical for deployment in real field environment.

## CONCLUSIONS

In this study, we improve the YOLOv8s model with lightweight by first replacing the C2f module in the YOLOv8s backbone network with the C-FasterNet module, secondly replacing the Conv in the backbone network with GhostConv, then adding the Focal CIoU to speed up the model convergence, and finally removing the large-target detection header in the model. This reduces the number of model parameters and computations while decreasing the model size and increasing the precision of the detection. The size of the improved lightweight model is only 52% of the original model; the number of parameters is only 51.4% of the original model; the computational volume is only 74.3% of the original model, and the precision of the model is improved by 0.2% compared with the original model, while the mAP@0.5:0.95 is improved by 0.269%. It shows that the improved model has high recognition accuracy while being lightweight, can accurately recognize overlapped and obscured wheat ears, and operates at a faster speed to meet the demand for real-time detection. This validates the feasibility of the improved lightweight model for wheat FHB in this study and provides a reference for the deployment of the model on mobile terminals in the next step.

## ACKNOWLEDGEMENTS

The authors are thankful to Guihong Yin and Guihua Bai for their strong support for this work. The authors would like to thank the editor and anonymous reviewers for their helpful comments and suggestions.

### Funding

This work was supported by the National Natural Science Foundation of China (No. 31501225), the Natural Science Foundation of Henan Province (No. 222300420463), the Natural Science Foundation of Henan Province (No. 232300420186), and the Joint Fund of Science and Technology Research and Development Plan of Henan Province (No. 222301420113). The funders had no role in study design, data collection and analysis, decision to publish, or preparation of the manuscript.

### Grant Disclosures

The following grant information was disclosed by the authors:
National Natural Science Foundation of China: 31501225.
Natural Science Foundation of Henan Province: 222300420463.
Natural Science Foundation of Henan Province: 232300420186.
Joint Fund of Science and Technology Research and Development Plan of Henan Province: 222301420113.

## Competing Interests

The authors declare that they have no competing interests.

## Author Contributions

- Chengkai Yang conceived and designed the experiments, performed the experiments, analyzed the data, performed the computation work, prepared figures and/or tables, authored or reviewed drafts of the article, and approved the final draft.
- Xiaoyun Sun analyzed the data, performed the computation work, prepared figures and/or tables, authored or reviewed drafts of the article, and approved the final draft.
- Jian Wang conceived and designed the experiments, prepared figures and/or tables, and approved the final draft.
- Haiyan Lv analyzed the data, authored or reviewed drafts of the article, and approved the final draft.
- Ping Dong conceived and designed the experiments, authored or reviewed drafts of the article, and approved the final draft.
- Lei Xi performed the experiments, authored or reviewed drafts of the article, and approved the final draft.
- Lei Shi conceived and designed the experiments, performed the experiments, analyzed the data, performed the computation work, prepared figures and/or tables, authored or reviewed drafts of the article, and approved the final draft.

## Data Availability

Manuscript in Supplemental File 1. Code and raw data in Supplemental File 2. Splitting data into data 1 and data 2 due to restrictions.

## Supplemental Information

Supplemental information for this article can be found online at http://dx.doi.org/10.7717/peerj-cs.1948#supplemental-information.

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
