# Peer review of "YOLOv8s-CGF: a lightweight model for wheat ear Fusarium head blight detection"

_PeerJ Computer Science, doi:10.7717/peerj-cs.1948_

## Round 0.1 · original submission · Major Revisions

This manuscript has been carefully examined by three experts in this field. The reviewers recognised some interesting findings and results but also raised reasonable comments and suggestions. Therefore, I would like the authors to carefully consider the reviewers' comments and make point-by-point revisions to improve the manuscript.

**Language Note:** The review process has identified that the English language must be improved. PeerJ can provide language editing services - please contact us at [email protected] for pricing (be sure to provide your manuscript number and title). Alternatively, you should make your own arrangements to improve the language quality and provide details in your response letter. – PeerJ Staff

Reviewer 1 ·

Basic reporting

Paper #93677: Chengkai Yang et al.'s study introduces the ‘YOLOv8s-CGF’ model for detecting Fusarium head blight (FHB) in wheat ears.

The manuscript was well written and structured.

It is an interesting study and it would have a nice group of readers in this journal.

Experimental design

The primary objective of this study is to streamline model parameters and reduce computing complexity while enhancing the accuracy of FHB detection in wheat. managers. A couple concerns that I think need to be addressed before publication, see below:

## Test of model performance
The modified model ‘YOLOv8s-CGF’ exhibits fewer parameters, reduced computational demands, and a smaller model size compared to original model ‘YOLOv8s’, shown in Table 2. It would be better for authors to present statistical tests among model performances.

## Classification of wheat FHB
While the model has a great performance in identifying the wheat FHB, it lacks the ability to classify severity of the disease. This classification is important for farmers to tailor appropriate treatment strategies. Different levels of severity may require different interventions, whether they involve fungicides, cultural practices, or other management techniques. Therefore, I suggest that authors might include a couple sentences in the discussion in this regard.

Validity of the findings

• In the introduction, the authors provided a review of previous studies on the performance of models such as 'YOLOv4' and 'YOLOv5' in detecting wheat FHB. However, there is a notable absence of information on studies related to the 'YOLOv8s' model for identifying wheat FHB. Considering that the proposed 'YOLOv8s-CGF' model is developed based on 'YOLOv8s,' I recommend that the authors include a brief review of previous studies specifically focused on the detection capabilities of 'YOLOv8s' in wheat FHB.
• Clarification is needed regarding whether the mentioned wheat in the paper is winter wheat or spring wheat.
• Page 1, lines 23-24: “The experimental results show that the size of the improved model (YOLOv8s-CGF) was only 11.7 M, which was 52.0% of the original model which is by YOLOv8s”. The sentence is not clear.
• Page 2, line 49, changing “adistance” to “distance”.
• Page 2, lines 59-61: “The current identification of FHB in wheat has, Hong et al. (2022) employed a lightweight improved YOLOv4-based FHB detection model for wheat ears, which achieved an accuracy of 93.69% for wheat ear FHB detection.”. Sentence is not clear, rephrase it.

Additional comments

It is a nice paper for being published

Cite this review as

Reviewer 2 ·

Basic reporting

Wheat Fusarium Head Blight is a serious disease and it can reduce production of wheat significantly. The research on how to quickly detect wheat Fusarium Head Blight is very meaningful.In this paper, a lightweight target detection model based on improved YOLOv8s is proposed to identify Wheat Fusarium Head Blight. The proposed model is innovative, and experimental results show that the model is very effective.
This paper can be accepted after minor revisions.
(1)There are a few grammar errors in this manuscript.
(2)The writing of the manuscript needs improvement with the assistance of English editing to improve readability.

Experimental design

no comment

Validity of the findings

no comment

Additional comments

no comment

Cite this review as

Reviewer 3 ·

Basic reporting

Please see the additional comments.

Experimental design

Please see the additional comments.

Validity of the findings

Please see the additional comments.

Additional comments

Wheat FHB is a kind of wheat disease that affects vast globally. Thus, timely and accurate identification of wheat FHB is crucial for improving wheat yield. This paper proposed the use of a YOLOv8-based method for wheat FHB detection. Compared with the original YOLOv8, a C-FasterNet module is proposed to replace the C2f module, GhostConv is used for Conv layers, A Focal CIoU is added, and the large-target detection headers are removed. The changes result in a computationally efficient YOLOv8 model for wheat FHB identification.

The manuscript is well-written in general. The research question is clearly defined. The experiments are well planned. And the evaluation shows a significant improvement from the baseline models. However, considering the application nature of the study, the reviewer suggests that the authors consider adding more introduction/discussion about the applicability.

For instance, how is the wheat FHB identified in the real world nowadays? More introduction and reference may be needed, especially for the recent development in the last few years. What are the limitations of the proposed method, and what are the future working directions? How can the algorithms be used in the real world? For example, can it be embedded in a drone for monitoring a large area?

Besides the discussion about the applicability, the language may need to be tuned. An English editing service may be needed.

Cite this review as

---

## Round 0.2 · accepted · Accept

The authors have addressed the comments and concerns raised by three expert reviewers. The manuscript has been revised to meet the publication standard of the journal. Therefore, I recommend an acceptance after English proofing.

Reviewer 1 ·

Basic reporting

Authors addressed my concerns well

Experimental design

A complete experimental setup and implementation.

Validity of the findings

The findings are publishable.

Additional comments

It should attract a wide range of readers for this paper

Cite this review as

Reviewer 2 ·

Basic reporting

The manuscript presents original research that significantly contributes to the field. The manuscript is well-written and easy to follow. The language is clear and concise, and the structure is logically organized.
Thus, I recommend acceptance of this manuscript for publication.

Experimental design

The experimental design of manuscript is sound, and the data collection and analysis procedures are clearly described.
The results are presented in a clear and logical manner, and the discussion section provides valuable insights into the implications of the findings. The manuscript also effectively compares and contrasts its results with those of previous studies.

Validity of the findings

The topic is well-chosen and the research question is both timely and relevant.The research methods are rigorous and appropriate for addressing the research question.

Cite this review as

Reviewer 3 ·

Basic reporting

The authors have addressed my concerns and I have no further questions. This article can be accepted for publication.

Experimental design

The authors have addressed my concerns and I have no further questions. This article can be accepted for publication.

Validity of the findings

The authors have addressed my concerns and I have no further questions. This article can be accepted for publication.

Cite this review as